# Commitment to protective measures during the COVID-19 pandemic in Syria: A nationwide cross-sectional study

Mosa Shibani[1][☼]*, Mhd Amin Alzabibi[1][☼], Abdul Fattah Mohandes[2], Humam Armashi[1], Tamim Alsuliman[3], Angie Mouki[4], Marah Mansour[5], Hlma Ismail[1], Shahd Alhayk[1], Ahmad abdulateef Rmman[1], Hala Adel Almohi Alsaid Mushaweh[2], Elias Battikh[6], Naram Khalayli[7], Bisher Sawaf[7], Mayssoun Kudsi[1,8]

1 Faculty of Medicine, Syrian Private University, Damascus, Syria, 2 Faculty of Medicine, University of Aleppo, Aleppo, Syria, 3 Hematology and Cell Therapy Department, Saint-Antoine Hospital, AP-HP Sorbonne University, Paris, France, 4 Faculty of Pharmacy, Maykop State Technological University, Maykop, Russia, 5 Faculty of Medicine, Tartous University, Tartous, Syria, 6 Department of Internal Medicine, Hamad Medical Corporation, Doha, Qatar, 7 Faculty of Medicine, Damascus University, Damascus, Syria, 8 Department of Rheumatology, Almowasa University Hospital, College of Medicine, Damascus University, Damascus, Syria

☼ These authors contributed equally to this work.
* Moosa.shibani@gmail.com

**Data Availability Statement:** All relevant data are within the paper and its Supporting information files.

## Abstract

### Background

Severe acute respiratory syndrome coronavirus 2 continues to impose itself on all populations of the world. Given the slow pace of vaccination in the developing world and the absence of effective treatments, adherence to precautionary infection control measures remains the best way to prevent the COVID-19 pandemic from spiraling out of control. In this study, we aim to evaluate the extent to which the Syrian population adheres to these measures and analyze the relationship between demographic variables and adherence.

### Methods

This cross-sectional study took place in Syria between January 17 and March 17, 2021. A structured self-administered questionnaire was used to collect the data. The questionnaire was distributed in both electronic and printed versions. Our sample consisted of 7531 individuals. Collected data were analyzed using SPSS v.25. The chi-square test was used to address the correlation between adherence and demographic variables.

### Results

Of the 10083 reached out, only 8083 responded, and 7531 included in the final analysis with an effective response rate of 74.7%. Of them, 4026 (53.5%) were women, 3984 (52.9%) were single, and 1908 (25.3%) had earned university degrees. 5286 (70.25) were in the high level of adherence category to protective measures. Statistically significant differences were documented when investigating the correlation between commitment to preventive measures and age, sex, marital status, financial status, employment, and educational

**Funding:** The author(s) received no specific funding for this work.

**Competing interests:** The authors have declared that no competing interests exist.

attainment. Furthermore, those who believed that COVID-19 poses a major risk to them, or society were more committed to preventive measures than those who did not.

## Conclusion

The participants in this study generally showed a high level of adherence to the preventive measures compared to participants in other studies from around the world, with some concerns regarding the sources of information they depend on. Nationwide awareness campaigns should be conducted and focus on maintaining, if not expanding, this level of commitment, which would mitigate the pandemic's impact on Syrian society.

## Background

The continued spread of severe acute respiratory syndrome coronavirus 2 (SARS-CoV-2)—the causative agent of coronavirus disease 2019 (COVID-19)—has impacted all aspects of life worldwide, with over 250 million cases and over 5 million deaths as of 13 December 2021 [1]. On 11 March 2020, the World Health Organization (WHO) officially declared COVID-19 to be a global pandemic [2], and recommended comprehensive strategies to prevent the spread of the virus [3]. Since person-to-person transmission mainly occurs via respiratory droplets, close contact with infected individuals during talking, sneezing, coughing, and indirect contact with contaminated objects or surfaces [4, 5], the most important recommendations include self-isolation, physical distancing, wearing face masks, and practicing hand hygiene [3, 6]. Many governments have implemented these virus-mitigation measures to contain the spread of the virus and protect vulnerable populations from infection. These cooperative efforts are important in lowering mortality rates and preventing health care systems from being overburdened. However, populations must be highly committed to these measures to ensure their success. In the absence of effective treatments and in light of recent evidence showing decreased vaccine-induced immunity after 5–7 months [7], preventive infection-control measures remain the best hope for containing the disease. The Syrian government began implementing precautionary measures to pre-empt the spread of the disease before the first case was even reported on 22 March 2020 [8, 9]. All schools, colleges, commercial and leisure centers, gyms, and places of worship were closed, and a 6 pm-to-6 am curfew was put into effect. However, these measures lasted only for two months (March to May 2020) [8]. A recent nationwide study in Syria reported good levels of awareness among the Syrian population regarding COVID-19 in general and preventive measures in particular [10]. Almost all participants (99%) were aware that proper hand hygiene, avoidance of crowded places, isolation at home, and wearing face masks in public places are the main preventive measures [10]. Information about the Syrian population's general knowledge regarding infection control will not only help to inform policy-makers as they make important decisions on how to best confront this pandemic, but it is also important to measure the extent to which the population adheres to these measures so that gaps between knowledge and practice can be addressed. As of 10 November 2021, 45,468 laboratory-confirmed cases and 2,637 casualties of COVID-19 have been reported by the Syrian Ministry of Health [11]. However, considering that testing has been limited in scale and that the cost of test kits is relatively high, it is likely that official numbers are deceptively low and do not reflect the severity of the pandemic in Syria. The war in Syria has raged for over 10 years and continues to impose massive burdens on the population, including economic, social, and educational challenges [12]. A consequence of this widespread conflict is

the largest refugee crisis since World War II [13]. The Syrian healthcare system, already devastated by the war and suffering staffing, supply, and funding shortages, has all but collapsed in the face of the pandemic. The Syrian economy has been particularly hard hit, and due to recent inflation, the number of people in need (PIN) is expected to increase from 11 million in 2020 to 13.3 million in 2021 [12]. The rising price of personal protective equipment and medical supplies, as well as other COVID-19 related factors that have increased the cost of living, has resulted in the majority of Syrian families being unable to afford leaving their jobs and sacrificing income to self-isolation and physical distancing measures [14]. In this study, we aim to measure the commitment of the Syrian population to infection prevention and control (IPC) measures (such as hand washing, wearing masks and gloves, and avoidance of handshaking and face-touching), measure perceived risk regarding COVID-19, and study the correlation between commitment and some demographic variables.

## Methods

### Study design, setting, and participants

A nationwide cross-sectional study was performed between January 17 to March 17, 2021. Data was collected using a structured self-administered questionnaire which was distributed to a sample of Syrian people. We developed the questionnaire based on previous studies and made some modifications to be suitable for Syrian society [15–17]. It was then piloted on a sample of 15 people to ensure clarity, and adjustments were made based on their feedback. Chain-referral (snowball sampling) and convenience sampling methods were employed by distributing the questionnaire in two formats: electronically as a Google Form survey via social media and messaging platforms (Facebook, Whatsapp, Twitter), and physically as hard copies to patients, their companions, and workers in public hospitals in each of Damascus, Aleppo, Homs, Tartous, Hama, and Sweida governorates. The sample size was calculated using Open-Epi online software available at "https://www.openepi.com/SampleSize/SSPropor.htm". According to data from the United Nations, the estimated population of Syria in 2019 was about 18 million [18]; based on this figure, the sample which is required to represent the total population was calculated to be at least 7336, with a confidence level of 95% and a confidence interval of 1.14.

Inclusion criteria were that the person is: (1) 18 years old or above, (2) literate, (3) a Syrian citizen living in Syria, and (4) willing to complete the questionnaire. In order to reach our desired sample size, initially, we reached out to 10083 individuals. Of them, 8083 agreed to participate. Of these 8083 participants, 551 were excluded for not meeting the inclusion criteria as follow: (17 withdrew their consent to participate, 543 were not Syrians or lived outside of Syria), yielding a final sample size of 7,531 responses which underwent statistical analysis.

### Measures

The questionnaire consisted of 32 questions divided into 3 sections:

1. *Socio-demographic characteristics*:
   13 questions about age, gender, marital status, nationality, governorate of origin, place of residence (urban or rural), financial status, employment status, educational level, father's and mother's educational level, health insurance coverage, and work or study in a health-care-related field. Financial status was asked as four categories: low, middle, good and Excellent. Age was divided into 4 groups: 18–24, 25–44, 45–65, and >65 years. Governorates were divided into 5 categories based on geographical location: 1- Central governorates (Damascus, Rif Dimashq, Hama, Homs), 2- Eastern governorates (Deir ez-Zor, Al-

Hasakah, Ar-Raqqah), 3- Western governorates (Latakia, Tartous), 4- Northern governorates (Aleppo, Idlib), 5-Southern governorates (Daraa, Quneitra, As-Suwayda).

2. *COVID-19 general information*:
   5 questions about prior infection with SARS-CoV-2, the risk this virus poses to the individual and to Syrian society as a whole, and participants' source of information.

3. *Commitment to infection prevention and control (IPC) measures*.
   12 yes-no statements about various IPC measures including: wearing a face mask, social event cancellation or postponement, self-isolation, cleaning or disinfecting touched items, carrying sanitizing hand-gel, reduced face-touching, healthy diet, avoiding people who have cold or flu-like symptoms, using tissues when sneezing or coughing, and washing hands with soap and water.

## Statistical analysis

Data from the hard copy questionnaires were entered manually by the investigators (MS, MAA, SA, and HI) to the original Google Forms online questionnaire that was used to collect online data, after which it was exported to a Microsoft Excel spreadsheet. The raw data was then encoded in Excel to make it compatible with the statistics software. A 13-point scale developed by the investigators was used to measure the level of commitment to IPC measures. Each individual measure was given one point (lowest = 0, highest = 13), then each participant was categorized into one of three categories based on how many protective measures he/she applied: 1- Low commitment (0–3 protective measures), 2- Moderate commitment (4–8 protective measures) and 3- High commitment (9–12 protective measures). We used Statistical Package for Social Sciences version 25.0 (SPSS Inc., Chicago, IL, United States) to analyze the data. Categorical variables were reported as frequencies and percentages. Pearson's chi-square test was used to study the associations between categorical groups. A $p$-value $< 0.05$ was considered statistically significant.

## Ethical considerations

The study protocol was approved by the respective Research Ethics Committee at each of Damascus, Aleppo, Tartous, and Syrian Private Universities, and the ethics committees of each hospital from which data was collected. Written informed consent was obtained from every participant as each questionnaire had an informed consent form (the first page in the hard copy version and the first question in the digital one) needs to be signed by the respondents prior to participation. All procedures performed in studies involving human participants were in accordance with the ethical standards of the institutional and/or national research committee and with the 1964 Helsinki declaration and its later amendments or comparable ethical standards.

## Results

### Participant characteristics

Of the 10083 person reached out, 8083 agreed to participate and 7531 fully completed the questionnaire and their responses were analyzed (effective response rate = 74.7%). Most of the respondents were females and the dominant age group was 18–24 years old. Over the half of the study population originated from the central governorates, and the lowest proportion were from the western governorates. Regarding marital status, over half of the participants were single. Most participants were financially middle class and only 371 (4.9%) had excellent financial

status. University students and university graduates represented the majority of responders. When asked if employed or studying in a healthcare-related field, over half of the participants answered "no". (Table 1).

### COVID-19 related information

When asked about a previous infection with COVID-19, the majority of participants answered "no", and only a small proportion had a PCR-confirmed infection. On the other hand, when the participants were asked if they know someone who has had a PCR-confirmed infection, most of them said "yes". Regarding the imposed risk of COVID-19 on Syrian society, more than half of the respondents believe it poses a major risk. However, when asked about the extent to which the virus poses a personal risk a considerably less proportion replied "major risk" (Table 2). Healthcare workers and social media (Facebook, Whatsapp, Youtube, Telegram, Instagram, etc.) were the main source of COVID-19 related information. (Fig 1).

### Commitment to preventive measures

The vast majority of participants showed good commitment (Table 3).

The most-practiced preventive measure among the study population was "covering the mouth/nose when coughing or sneezing" followed by "hand washing with soap and water more often than usual". Half of the study population started to follow a healthy diet and over half of them reduced their attendance at school, college, university or work. (Table 4).

### Correlations between commitment to preventive measures and participants characteristics

Chi-square univariate analysis showed a statistically significant difference between males and females regarding commitment to preventive measures. 76.4% of females and 63.1% of males were categorized as highly committed, while only 3.4% of females and 6.2% of males were categorized as poorly committed ($\chi^2$ = 160.683, $p$-value<0.001). A significant association was found between age groups and adherence to preventive measures: 25–44 year old participants were most committed to IPC measures with 1780(76.1%) categorized as 'high', followed by 18–24, 45–65 and >65 age groups: 2269(72.6%), 1022(60.6%) and 215(56.1%) respectively ($\chi^2$ = 204.974, $p$-value <0.001). Participants in relationships (76.6%, $\chi^2$ = 92.002, $p$-value <0.001) and those from the western governorates of Syria (78.8%, $\chi^2$ = 184.079, $p$-value <0.001) were more committed to IPC measures than their single counterparts and those in other parts of the country. Commitment to preventive measures was significantly associated with residency and financial status, with urban residents (71.6% vs. 35.4%, $\chi^2$ = 59.106, $p$-value <0.001) and those in good financial status (76.6%, $\chi^2$ = 279.195, $p$-value <0.001) were the most committed groups. Participants with post-graduate education (80.6%, $\chi^2$ = 640.976, $p$-value <0.001) and students with full time jobs (77.1%, $\chi^2$ = 129.431, $p$-value <0.001) were the most committed to preventive measures. (Table 5) Our results revealed that people who believe that COVID-19 poses a major risk to Syrian society were more committed to IPC measures, with 77.4% being highly committed compared to 59.5% from the 'minor risk' group and 43.0% from the 'no risk at all' group. Similarly, those who believe that COVID-19 poses a major risk to them personally were more committed to preventive measures, as (79.5%) of them were in the high commitment category. (Table 6).

### Discussion

The COVID-19 pandemic has significantly impacted humanity and forced governments across the world to adopt extensive infection prevention and control measures with varying

**Table 1. Participant's characteristics.**

| Variables | | Total (%) |
|---|---|---|
| | | **n = 7531** |
| Age range (years) | 18–24 | 3124 (41.5%) |
| | 25–44 | 2338 (31%) |
| | 45–65 | 1686 (22.4%) |
| | > 65 | 383 (5.1%) |
| Gender | Male | 3505 (46.5%) |
| | Female | 4026 (53.5%) |
| Marital status | Single | 3984 (52.9%) |
| | Married | 2825 (37.5%) |
| | In relationship | 500 (6.6%) |
| | Widow | 222 (2.9%) |
| Do you have health insurance? | Yes | 1489 (19.8%) |
| | No | 6042 (80.2%) |
| Educational level | No formal education | 324 (4.3%) |
| | Elementary school | 422 (5.6%) |
| | Secondary school | 550 (7.3%) |
| | Highschool | 782 (10.4%) |
| | University student | 2906 (38.6%) |
| | University graduate | 1908 (25.3%) |
| | Postgraduate degree | 639 (8.5%) |
| Do you work or study in the healthcare system | Yes | 2595 (34.5%) |
| | No | 4936 (65.5%) |
| Mother's educational level | No formal education | 1225 (16.3%) |
| | Primary school | 1834 (24.4%) |
| | Secondary school | 1656 (22%) |
| | University degree | 2625 (34.9%) |
| | Postgraduate degree | 191 (2.5%) |
| Father's educational level | No formal education | 760 (10.1%) |
| | Primary school | 1971 (26.2%) |
| | Secondary school | 1512 (20.1%) |
| | University degree | 2776 (36.9%) |
| | Postgraduate degree | 512 (6.8%) |
| Residency | City | 5711 (75.8%) |
| | Countryside | 1820 (24.2%) |
| Geographical origin | Eastern Syria | 365 (4.8%) |
| | Northern Syria | 1272 (16.9%) |
| | Middle Syria | 4376 (58.1%) |
| | Southern Syria | 353 (4.7%) |
| | Western Syria | 1165 (15.5%) |
| Financial status | Low | 1268 (16.8%) |
| | Middle | 3241 (43%) |
| | Good | 2651 (35.2%) |
| | Excellent | 371 (4.9%) |

(*Continued*)

**Table 1.** (Continued)

| Variables | | Total (%) |
|---|---|---|
| | | **n = 7531** |
| Employment | Unemployed | 1635 (21.7%) |
| | Part-time job | 1114 (14.8%) |
| | Full-time job | 1688 (22.4%) |
| | Student | 2653 (35.2%) |
| | Student + Part-time job | 290 (3.9%) |
| | Student + Full-time job | 131 (1.7%) |
| | Part-time + Full-time jobs | 20 (0.3%) |

degrees of severity [19]. Therefore most of the COVID-19 cases are asymptomatic or very mild. Syria is a low-income country that has been ravaged by civil war for over a decade, diminishing the ability to adequately respond to the pandemic and impose meaningful quarantines. As such, results of measures adopted by other countries and regions cannot be relied upon to predict the course of the pandemic in Syria, and the extraordinary difficulties facing the country and the realities on the ground must be taken into account when uniquely assessing the situation in Syria. The efficiency and impact of infection prevention and control (IPC) measures can be optimized by obtaining insights into the population's current commitment to such measures. To the best of our knowledge, this is the first nationally representative study to offer insights into people's adherence to preventive measures during the COVID-19 pandemic in Syria. Our findings revealed that 70.2% of the population claim to adhere to most of the preventive measures asked about in the questionnaire. This level of adherence is similar to that found in a Belgian study, and better than that from an Ethiopian study [20, 21]. The majority of our participants were young, and the age distribution of our population was generally

**Table 2. COVID-19 related information.**

| | | N (%) |
|---|---|---|
| Do you believe you have had coronavirus? | No | 4549 (60.4) |
| | Yes, with PCR | 240 (3.2) |
| | Yes, with symptoms | 2742 (36.4) |
| Do you personally know anyone (excluding yourself) who has had a PCR-confirmed COVID-19? | No | 2960 (39.3) |
| | Yes | 4571 (60.7) |
| To what extent do you think COVID-19 poses a risk to people in Syria? | No risk at all | 314 (4.2) |
| | Minor risk | 1621 (21.5) |
| | Major risk | 464 (61.6) |
| | Do not know | 956 (12.7) |
| To what extent do you think COVID-19 poses a risk to you personally? | No risk at all | 834 (11.1) |
| | Minor risk | 3224 (42.8) |
| | Major risk | 2218 (29.5) |
| | Do not know | 1255 (16.7) |

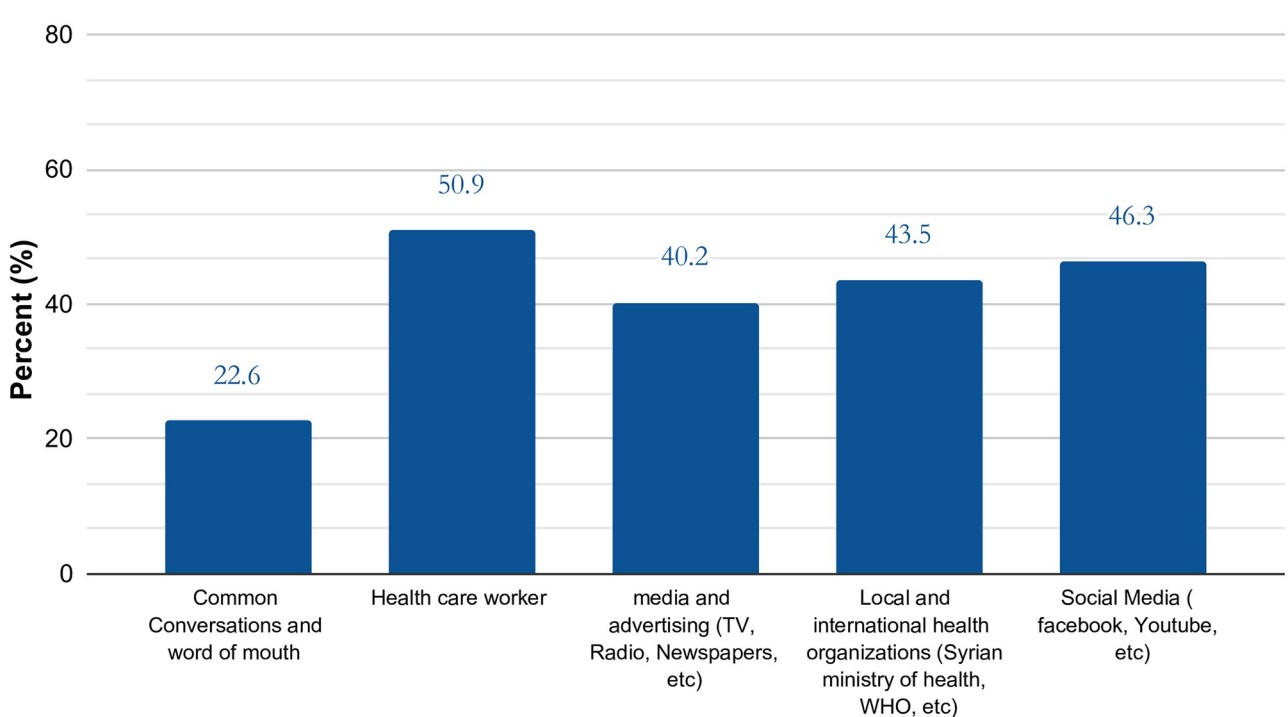

**Fig 1. Participants sources of information.**

consistent with the demographic data reported by the Central Bureau of Statistics (CBS), Damascus, Syria [22]. According to the latest CBS report, 40% of the Syrian population were below 24 years old, and 25.5% were 25–44 years old (compared to 41.5% and 31%, respectively, of our study population) [22]. The importance of face masks in reducing the spread of the virus is supported by numerous studies [23, 24]. One study suggests that complete eradication of the disease can be achieved if 80% of the population uses face masks effectively [25]. The vast majority of our participants (87.3%) were committed to wearing face masks in public

**Table 3. Level of commitment to protective measures.**

| The number of preventive measures used by the participants | | | |
|---|---|---|---|
| Number of measures | N(%) | Level of commitment | N(%) |
| 0 | 91(1.2%) | Low commitment | 357(4.7%) |
| 1 | 67(0.9%) | | |
| 2 | 93(1.2%) | | |
| 3 | 106(1.4%) | | |
| 4 | 159(2.1%) | Moderate commitment | 1888(25.1%) |
| 5 | 221(2.9%) | | |
| 6 | 328(4.4%) | | |
| 7 | 461(6.1%) | | |
| 8 | 719(9.5%) | | |
| 9 | 932(12.4%) | High commitment | 5286(70.2%) |
| 10 | 1278(17.0%) | | |
| 11 | 1511(20.1%) | | |
| 12 | 1565(20.8%) | | |

**Table 4. Specific preventive measures.**

| | Yes (N%) | No (N%) |
|---|---|---|
| Wore a face mask when in crowded places or when public places | 6577 (87.3) | 954 (12.7) |
| Reduced the amount you go to school, college, university or work | 4950 (65.7) | 2581 (34.3) |
| Cancelled or postponed a social event such as meeting friends, eating out or going to a sporting event | 5508 (73.1) | 2023 (26.9) |
| Reduced the number of times you go to shops | 5764 (76.5) | 1767 (23.5) |
| Kept away from crowded places? | 6317 (83.9) | 1214 (16.1) |
| Cleaned or disinfected things you might touch (doorknob or hard surfaces) | 5517 (73.3) | 2014 (26.7) |
| Carried sanitizing hand gel with you when you were out? | 5557 (73.8) | 1974 (26.2) |
| Reduced the amount you touch your eyes, nose, and\or mouth? | 5722 (76) | 1809 (24) |
| Followed a healthy diet or took vitamins supplements | 3747 (49.8) | 3784 (50.2) |
| Tried to avoid people who have cold or flu-like symptoms? | 6645 (88.2) | 886 (11.8) |
| Usually used tissues when sneezing or coughing | 6887 (91.4) | 644 (8.6) |
| Washed your hands with soap and water more often than usual? | 6637 (88.1) | 894 (11.9) |

spaces. This proportion is predictably lower than those from studies in China (98.0%) and Hong Kong (98.8%), where mask-wearing has been ingrained in the culture for decades, but considerably higher than in studies from Northwest Ethiopia (32.42%), Ethiopia (13.9%), Saudi Arabia (56.4%), and the United Kingdom (3.1%) [15, 21, 26–28].

Low income and unemployment were correlated with lower adherence to IPC measures, while higher income and gainful employment was correlated with higher adherence; only 55% of low income responders were highly committed to IPC measures, compared to 76.6% of those with good financial status. This may owe to the high cost of commitment to protective measures, which is prohibitively expensive for a significant proportion of the Syrian population. The Syrian pound has lost 35% of its value against the US dollar in the last year alone [14], and the percentage of the population living in poverty is 90% and rising [29]. While screening tests and social distancing might be considered cost-effective elsewhere in the world, this is not the case for the Syrian population (PCR screens cost $50 each and are not subsidized). Infection control measures such as sweeping lockdowns have only recently affected most of the world's economies, whereas the Syrian economy, already suffering from a decade of war and crippling sanctions, has been devastated by lockdowns and other pandemic-related economic pressures. As such, Syria's population and healthcare system are in desperate need of international support in the form of financial grants and donations of personal protective equipment, drugs, medical supplies, and vaccines. Income and employment-related results are in line with studies from China and Ethiopia, but do not align with those of a study from Saudi Arabia, likely due to vastly different socio-economic dynamics [15, 26, 28].

UNICEF reported that after a decade of war in Syria, more than half of children continue to be deprived of education [30]. The enormous scale of the education crisis is extremely worrying, as it threatens not only the future of an entire generation of children and the country as a

**Table 5. Correlation between demographic characteristics and level of commitment to protective measures.**

| Demographic characteristics | | Level of commitment | | | Chi-Square value | P-value |
|---|---|---|---|---|---|---|
| | | Low | Moderate | High | | |
| Age (years) | 18–24 (n = 3124) | 82 (2.6%) | 773(24.7%) | 2269(72.6%) | 204.974 | <0.001* |
| | 25–44 (n = 2338) | 101(4.3%) | 457(19.5%) | 1780(76.1%) | | |
| | 45–65 (n = 1686) | 137(8.1%) | 527(31.3%) | 1022(60.6%) | | |
| | > 65 (n = 383) | 37(9.7%) | 131(34.2%) | 215(56.1%) | | |
| Gender | Male (n = 3505) | 219(6.2%) | 1075(30.7%) | 2211(63.1%) | 160.683 | <0.001* |
| | Female (n = 4026) | 138(3.4%) | 813(20.2%) | 3075(76.4%) | | |
| marital status | Single (n = 3984) | 134(3.4%) | 978(24.5%) | 2872(72.1%) | 92.002 | <0.001* |
| | Married (n = 2825) | 202(7.2%) | 719(25.5%) | 1904(67.4%) | | |
| | in relationship (n = 500) | 6(1.2%) | 111(22.2%) | 383(76.6%) | | |
| | Widow (n = 222) | 15(6.8%) | 80(36.0%) | 127(57.2%) | | |
| origin | Eastern (n = 365) | 25(6.8%) | 121(33.2%) | 219(60.0%) | 184.079 | <0.001* |
| | Northern (n = 1272) | 128(10.1%) | 384(30.2%) | 760(59.7%) | | |
| | Middle (n = 4376) | 164(3.7%) | 1072(24.5%) | 3140(71.8%) | | |
| | Southern (n = 353) | 12(3.4%) | 92(26.1%) | 249(70.5%) | | |
| | Western (n = 1165) | 28(2.4%) | 219(18.8%) | 918(78.8%) | | |
| residency | City (n = 5711) | 213(3.7%) | 1409(24.7%) | 4089(71.6%) | 59.106 | <0.001* |
| | Countryside (n = 1820) | 144(7.9%) | 479(26.3%) | 1197(65.8%) | | |
| financial status | Low (n = 1268) | 150(11.8%) | 420(33.1%) | 698(55.0%) | 279.195 | <0.001* |
| | Middle (n = 3241) | 134(4.1%) | 814(25.1%) | 2293(70.7%) | | |
| | Good (n = 2654) | 59(2.2%) | 561(21.2%) | 2031(76.6%) | | |
| | Excellent (n = 371) | 14(3.8%) | 93(25.1%) | 264(71.2%) | | |
| employment | Unemployed (n = 1635) | 148(9.1%) | 427(26.1%) | 1060(64.8%) | 129.431 | <0.001* |
| | part-time (n = 1114) | 75(6.7%) | 274(24.6%) | 765(68.7%) | | |
| | full-time (n = 1688) | 60(3.6%) | 404(23.9%) | 1224(72.5%) | | |
| | Student (n = 2653) | 64(2.4%) | 675(25.4%) | 1914(72.1%) | | |
| | part-time + student (n = 290) | 5(1.7%) | 77(26.6%) | 208(71.7%) | | |
| | full-time + student (n = 131) | 4(3.1%) | 26(19.8%) | 101(77.1%) | | |
| | part-time + full-time(n = 20) | 1(5.0%) | 5(25.0%) | 14(70.0%) | | |
| academic level | no education (n = 324) | 81(25.0%) | 129(39.8%) | 114(35.2%) | 640.976 | <0.001* |
| | Elementary (n = 422) | 61(14.5%) | 151(35.8%) | 210(49.8%) | | |
| | Secondary (n = 550) | 35(6.4%) | 182(33.1%) | 333(60.5%) | | |
| | High school (n = 782) | 40(5.1%) | 212(27.1%) | 530(67.8%) | | |
| | university student (n = 2906) | 78(2.7%) | 721(24.8%) | 2107(72.5%) | | |
| | university graduate (n = 1908) | 39(2.0%) | 392(20.5%) | 1477(77.4%) | | |
| | post-university study (n = 639) | 23(3.6%) | 101(15.8%) | 515(80.6%) | | |
| father's educational level | no education (n = 760) | 142(18.7%) | 257(33.8%) | 361(47.5%) | 482.949 | <0.001* |
| | primary education (n = 1971) | 95(4.8%) | 543(27.5%) | 1333(67.6%) | | |
| | secondary education (n = 1521) | 40(2.6%) | 368(24.3%) | 1104(73.0%) | | |
| | university degree (n = 2776) | 69(2.5%) | 628(22.6%) | 2079(74.9%) | | |
| | post- university (n = 512) | 11(2.1%) | 92(18.0%) | 409(79.9%) | | |
| mother's educational level | no education (n = 1225) | 170(13.9%) | 408(33.3%) | 647(52.8%) | 391.734 | <0.001* |
| | primary education (n = 1834) | 72(3.9%) | 494(26.9%) | 1268(69.1%) | | |
| | secondary education (n = 1656) | 52(3.1%) | 391(23.6%) | 1213(73.2%) | | |
| | university degree (n = 2625) | 58(2.2%) | 560(21.3%) | 2007(76.5%) | | |
| | post-university (191) | 5(2.6%) | 35(18.3%) | 151(79.1%) | | |

* P-value<0.05 considered statistically significant

**Table 6. Correlation between risk perception and commitment to preventive measures.**

| | | Level of commitment | | | Chi-Square | p.value |
|---|---|---|---|---|---|---|
| | | **Low** | **Moderate** | **Good** | | |
| To what extent do you think coronavirus poses a risk to people in Syria? | No risk at all | 54(17.2%) | 125(39.8%) | 135(43.0%) | 433.437 | <0.001* |
| | Minor risk | 126(7.8%) | 531(32.8%) | 964(59.5%) | | |
| | Major risk | 102(2.2%) | 947(20.4%) | 3591(77.4%) | | |
| | Do not know | 75(7.8%) | 285(29.8%) | 596(62.3%) | | |
| To what extent do you think coronavirus poses a risk to you personally? | No risk at all | 119(14.3%) | 311(37.3%) | 404(48.4%) | 402.271 | <0.001* |
| | Minor risk | 98(3.0%) | 858(26.6%) | 2268(70.3%) | | |
| | Major risk | 50(2.3%) | 405(18.3%) | 1763(79.5%) | | |
| | Do not know | 90(7.2%) | 314(25.0%) | 851(67.8%) | | |

* P-value<0.05 considered statistically significant

whole, but also the important role of schools as conduits for health literacy and education about diseases and the importance of infection control. Our study revealed an important correlation between education and adherence to IPC measures, with commitment increasing significantly as the level of education increases. On one end of the spectrum, only 35.2% of uneducated participants adhere to protective measures, compared to 80.6% of participants with postgraduate education on the other end. These findings are consistent with studies from China, Ethiopia, and Germany [16, 26, 28].

Numerous studies have shown that risk perception can be considered a determinant of individual behavior during a disease outbreak [31–33]. The more risk perceived, the more likely people are to adhere to preventive measures. Some studies go even further and suggest that it is important to differentiate between the 'experiential' and 'affective' components of risk perception [34, 35]. Earlier research demonstrated that experiential risk perception, "the gut feeling of being vulnerable to risk", was positively associated with applying personal protective actions, such as vaccination and sun protection [36, 37]. Our study seems to support this assumption, and other studies in Italy, Northwest Ethiopia reported the same observation [21, 38].

Several previous studies showed that the level of knowledge correlates directly with adherence to preventive measures [28, 39]. Ideally, the public should be well-informed by reliable sources of information. Unfortunately, our participants' reliance on untrusted sources on social media is part of a global trend in which misinformation is rampant and pervasive. Social media tends to be the most expedient means of obtaining information for many people, and studies have shown that social media is a fertile and target-rich environment for spreading misinformation and conspiracy theories that negatively affect the quality of the public's knowledge [40–42]. Since it is impossible to fully control what is published on social media, local and global health authorities must enhance their presence on these platforms and use engaging content and effective methods to spread awareness and accurate information. Studying the public's perception and behavior toward COVID-19 provides valuable insight which can help policymakers and healthcare providers to address the knowledge gaps that negatively affect people's perception and behavior, thereby improving the national response to this pandemic. We encourage all concerned institutions to invest the time, resources, and expertise necessary to successfully and significantly leverage social media platforms to drive public health education and COVID-19 awareness campaigns. The rebuilding and rehabilitation of schools must be prioritized, and infection control measures incorporated into the curriculum. Special accommodations should be made for low-income people and families, in the form of

distributing infection control kits (composed of a reusable face mask and hand sanitizer) and securing their income when proven to be sick to encourage them to self-quarantine.

## Conclusion

Despite the high level of commitment to infection prevention and control (IPC) measures demonstrated by the participants in our study, it is necessary to stress the importance of continuing this commitment throughout the pandemic. It is recommended that local and international health authorities carry out continuous awareness campaigns with the aim of reminding the population of the importance of consistently applying IPC measures. Moreover, the population should be educated about how to identify and avoid misinformation on social media and to rely on reliable sources of information. Because of the economic and humanitarian situation in war-torn Syria, it is necessary for all concerned bodies and organizations to take serious action and provide appropriate assistance to the healthcare system to help contain this pandemic.

## Limitations

This study is subject to some limitations. First, as a cross-sectional study it may not be able to determine causation, therefore more longitudinal studies are recommended. Second, distributing the questionnaire online only will lead to selection bias, as most people with internet access tend to be younger and wealthier. To minimize this bias we distributed the questionnaire both online and as hard copies. Third, many questions were subject to recall bias. Finally, the economic status question was subjective since the value of the Syrian pound is not stable and the exchange rate continues to fluctuate. This continues to affect the purchasing power of the local currency, with many families whose income was once adequate falling below the poverty line.

## Supporting information

**S1 Data.**
(XLSX)

## Acknowledgments

We would like to express our thanks to all the people and organizations that helped in distributing this survey, especially: Syrian Researchers Organization, Impact Makers Team, Syrian Drugs Up To Date Platform, MedDose Organization (NGO). We would also like to acknowledge the valuable help provided by Dr. Dana Alakhrass in revising this manuscript.

## Author Contributions

**Conceptualization:** Mosa Shibani, Mhd Amin Alzabibi, Tamim Alsuliman, Hlma Ismail, Bisher Sawaf, Mayssoun Kudsi.

**Data curation:** Abdul Fattah Mohandes, Humam Armashi, Marah Mansour, Hlma Ismail, Shahd Alhayk, Ahmad abdulateef Rmman, Hala Adel Almohi Alsaid Mushaweh, Elias Battikh, Naram Khalayli.

**Formal analysis:** Tamim Alsuliman.

**Investigation:** Naram Khalayli, Bisher Sawaf.

**Methodology:** Mosa Shibani, Mhd Amin Alzabibi, Abdul Fattah Mohandes, Angie Mouki, Marah Mansour, Shahd Alhayk, Ahmad abdulateef Rmman, Hala Adel Almohi Alsaid Mushaweh, Elias Battikh, Naram Khalayli, Bisher Sawaf.

**Resources:** Mosa Shibani, Mhd Amin Alzabibi, Abdul Fattah Mohandes, Humam Armashi, Shahd Alhayk, Ahmad abdulateef Rmman, Hala Adel Almohi Alsaid Mushaweh, Elias Battikh, Naram Khalayli, Bisher Sawaf.

**Software:** Naram Khalayli.

**Supervision:** Mosa Shibani, Mhd Amin Alzabibi, Tamim Alsuliman, Bisher Sawaf, Mayssoun Kudsi.

**Writing – original draft:** Mosa Shibani, Mhd Amin Alzabibi, Abdul Fattah Mohandes, Hlma Ismail, Shahd Alhayk.

**Writing – review & editing:** Humam Armashi, Tamim Alsuliman, Angie Mouki, Bisher Sawaf, Mayssoun Kudsi.

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
