## [Decision Letter · Decision Letter 0]

6 Jan 2022

PONE-D-21-39441Commitment to protective measures during COVID-19 pandemic in Syria: a nationwide cross-sectional study.PLOS ONE

Dear Dr. Shibani,

Thank you for submitting your manuscript to PLOS ONE. After careful consideration, we feel that it has merit but does not fully meet PLOS ONE’s publication criteria as it currently stands. Therefore, we invite you to submit a revised version of the manuscript that addresses the points raised during the review process.

We look forward to receiving your revised manuscript.

Kind regards,

Sanjay Kumar Singh Patel, Ph.D.

Academic Editor

PLOS ONE

Journal Requirements:

Reviewers' comments:

Reviewer's Responses to Questions

**Comments to the Author**

1. Is the manuscript technically sound, and do the data support the conclusions?

Reviewer #1: Yes

Reviewer #2: Yes

Reviewer #3: Yes

2. Has the statistical analysis been performed appropriately and rigorously? 

Reviewer #1: Yes

Reviewer #2: Yes

Reviewer #3: Yes

3. Have the authors made all data underlying the findings in their manuscript fully available?

Reviewer #1: Yes

Reviewer #2: Yes

Reviewer #3: Yes

4. Is the manuscript presented in an intelligible fashion and written in standard English?

Reviewer #1: Yes

Reviewer #2: Yes

Reviewer #3: Yes

5. Review Comments to the Author

Reviewer #1: In this paper entitled "Commitment to protective measures during COVID-19 pandemic in Syria: a nationwide cross-sectional study", the authors evaluate the extent to which the Syrian population adheres to these measures and analyze the relationship between demographic variables and adherence. The manuscript is well written and intelligently presented. The study is nicely designed, and the sample size is calculated using OpenEpi online software. All the data is analyzed using Statistical Package for Social Sciences version 25.0. The results are presented in a tabular form that gives a complete overview of the data collected. In addition, percentage, correlation and p-value are calculated to provide statistical stringency to the manuscript. I congratulate the authors for this work and recommend this work for publication. The manuscript is suitable for publications and should be formally accepted once it meets all technical requirements. Although it has one or two typos in the manuscript (lines 229,230),it needs to provide high-resolution fig 1 or remove it from the manuscript.

Reviewer #2: The manuscript entitled “Commitment to protective measures during COVID-19 pandemic in Syria: a nationwide cross-sectional study.” written by Shibani et al investigated the extent to which the Syrian population adheres to precautionary infection control measures and analyze the relationship between demographic variables and adherence. Authors did a cross-sectional study in Syria between January 17 and March 17, 2021 using a structured self-administered questionnaire. Authors concluded that this study generally showed a high level of adherence to the preventive measures compared to participants in other studies from around the world, with some concerns regarding risk perception and the sources of information they depend on. This manuscript requires minor revision prior to its publication in PLOS One as follows:

Suggestions: -

1. The background section can be updated with minor information such as health, diet and immunity, the natural COVID-19 preventive measures and various variants of COVID-19 i.e. doi: 10.1007/s12088-020-00893-4;  doi: 10.1007/s12088-020-00908-0; doi: 10.1007/s15010-021-01734-2.

2. Figures quality may be improved with high resolution images (minor).

3. It would be required to provide one illustrative Figure as to highlight the summary or future prospect of this study.

4. Did authors ask about vaccination status in their survey (vaccinated/not vaccinated/will receive vaccination/reluctant for vaccination?

5. The English of manuscript can be polished (minor).

6. The authors should cross-check all abbreviations in the manuscript. Initially, define in full name followed by abbreviation.

Reviewer #3: This study investigated the commitment status of Syrian population to the protective measures against SARS-CoV-2 viral infection during COVID-19 pandemic. The manuscript is well organized and worth to be published. Yet, given the high contagious nature of the SARS-CoV-2 virus, commitment to the protective measures may only help to delay the infection to some limited extent, and eventually all members of the public will get infected by the virus in the long time. So, the paper will be more attractive if the authors can also provide some insightful understanding about infectious diseases like the SARS-CoV-2 infection, so that the public know what to do to avoid to develop into severe case of COVID-19 if they are infected by SARS-CoV-2 virus.

In talking about infectious diseases, it would be better if we distinguish infections from diseases. Infection is the presence of pathogen, which is not a sufficient condition for disease. [1]

Like most of the other viral and bacterial infections, the SARS-COV-2 viral infection is self-limiting [2]. When the human cells are infected by virus or bacteria, the human immunity will actively induce cell self-destruction (programmed cell deaths like necroptosis [3] and pyroptosis [4]) to stop the intracellular infection and reuse the nutrition from the degradation of these destroyed cells and the microorganisms in these cells to rebuild the tissue cells [5]. So by actively destroy damaged somatic cells, the viral and bacterial infections are also removed and become self-limiting [6]. Therefore most of the COVID-19 cases are asymptomatic or very mild.

Most of the severe cases of infectious diseases are the result of the overreaction of the strong yet malfunctioning immune system, as Sir William Osler stated more than 100 years ago: "Except on few occasions, the patient appears to die from the body’s response to infection rather than from it. Sir William Osler (1904)" [7]. As the human immune system also plays a virtual role in nutrition acquisition from the degradation of viral damaged epithelial cells, the nutrition surge coupled with the overnutrition state in some patients with obesity or metabolic syndromes may contribute to lipotoxicity and damage in non-adipose tissues, triggering hyperinflammation and contribute to the malfunction of the immune system. So the SARS-CoV-2 viral infection is only the trigger for the COVID-19 disease, and the real cause for the severe cases of COVID-19 disease are autoimmune disorder [8] caused by overnutrition which leads to the hyperinflammation and cytokine storm in these patients.

References:

1. V Humphries DL, Scott ME, Vermund SH (2021) Pathways linking nutritional status and infectious disease: causal and conceptual frameworks. In: Nutrition and infectious diseases, Shifting the Clinical Paradigm (eds DL Humphries, ME Scott, SH Vermund), pp. 3–22. Cham, Switzerland: Humana. DOI: 10.1007/978-3-030-56913-6_1

2. Kumar S, Veldhuis A, Malhotra T (2021) Neuropsychiatric and Cognitive Sequelae of COVID-19. Front Psychol 12:577529. DOI: 10.3389/fpsyg.2021.577529.

3. Nailwal, H., Chan, F.KM. Necroptosis in anti-viral inflammation. Cell Death Differ 26, 4–13 (2019). DOI: 10.1038/s41418-018-0172-x

4. Jorgensen I, Miao EA (2015) Pyroptotic cell death defends against intracellular pathogens. Immunological Reviews. 265 (1): 130-142. DOI:10.1111/imr.12287

5. Maitre Y, Mahalli R, Micheneau P, Delpierre A, Amador G, Denis F (2021) Evidence and Therapeutic Perspectives in the Relationship between the Oral Microbiome and Alzheimer's Disease: A Systematic Review. Int J Environ Res Public Health 18(21):11157. DOI: 10.3390/ijerph182111157.

6. Levin BR, Baquero F, Ankomah P, McCall IC (2017) Phagocytes, Antibiotics, and Self-Limiting Bacterial Infections. Trends in Microbiology, 25(11):878-892. DOI: 10.1016/j.tim.2017.07.005

7. Dobson GP, Biros E, Letson HL and Morris JL (2021) Living in a Hostile World: Inflammation, New Drug Development, and Coronavirus. Front. Immunol. 11:610131. DOI: 10.3389/fimmu.2020.610131

8. Halpert G, Yehuda Shoenfeld Y (2020) SARS-CoV-2, the autoimmune virus. Autoimmunity Reviews 19(12):102695. DOI: 10.1016/j.autrev.2020.102695.

---

## [Author Response · Author response to Decision Letter 0]

27 Feb 2022

Response letter

 We would like to thank the editor and reviewers for taking their valuable time to review this manuscript. We have thoroughly assessed the comments and implemented them into our revised manuscript, now that this article has been made suitable we aspire to have this article published in your journal. 

The following letter will report the responses to the reviewers’ comments.

Journal Requirements:

Author response: All style requirements have been addressed, including those for file naming.

Author response: We apologize for this mistake and have deleted The ethical statement from the other sections.

Author response: The reference list have been revised thoroughly. 

Review Comments to the Author

Reviewer #1: In this paper entitled "Commitment to protective measures during COVID-19 pandemic in Syria: a nationwide cross-sectional study", the authors evaluate the extent to which the Syrian population adheres to these measures and analyze the relationship between demographic variables and adherence. The manuscript is well written and intelligently presented. The study is nicely designed, and the sample size is calculated using OpenEpi online software. All the data is analyzed using Statistical Package for Social Sciences version 25.0. The results are presented in a tabular form that gives a complete overview of the data collected. In addition, percentage, correlation and p-value are calculated to provide statistical stringency to the manuscript. I congratulate the authors for this work and recommend this work for publication. The manuscript is suitable for publications and should be formally accepted once it meets all technical requirements. Although it has one or two typos in the manuscript (lines 229,230),it needs to provide high-resolution fig 1 or remove it from the manuscript.

Author response: We apologize for this mistake and have revised this manuscript thoroughly for any other typos. We also provided a high-resolution copy of figure 1.

Reviewer #2: The manuscript entitled “Commitment to protective measures during COVID-19 pandemic in Syria: a nationwide cross-sectional study.” written by Shibani et al investigated the extent to which the Syrian population adheres to precautionary infection control measures and analyze the relationship between demographic variables and adherence. Authors did a cross-sectional study in Syria between January 17 and March 17, 2021 using a structured self-administered questionnaire. Authors concluded that this study generally showed a high level of adherence to the preventive measures compared to participants in other studies from around the world, with some concerns regarding risk perception and the sources of information they depend on. This manuscript requires minor revision prior to its publication in PLOS One as follows:

Suggestions: 

1. The background section can be updated with minor information such as health, diet and immunity, the natural COVID-19 preventive measures and various variants of COVID-19 i.e. doi: 10.1007/s12088-020-00893-4; doi: 10.1007/s12088-020-00908-0; doi: 10.1007/s15010-021-01734-2.

Author response: We thank the reviewer for the guidance, and have used the valuable information provided above to enhance our paper quality. These amendments have been made to the manuscript text and where these can be viewed (Background section, lines 67-68)

2. Figures quality may be improved with high resolution images (minor).

Author response: We apologize for the inconvenience and have provided a high-resolution copy of figure 1.

3. It would be required to provide one illustrative Figure as to highlight the summary or future prospect of this study.

Author response: We thank the reviewer for this comment, and would like to ask for further instructions about how to address this point.

4. Did authors ask about vaccination status in their survey (vaccinated/not vaccinated/will receive vaccination/reluctant for vaccination?

Author response: We thank the reviewer for this comment, and would like to clarify that at the time when this study was conducted, no vaccine were available in Syria, so our main focus was to study the commitment to personal protective measures regardless of the availability of the vaccines. However, there is another study was conducted at the same period regarding this manner i.e.: https://doi.org/10.1186/s12889-021-12186-6

5. The English of manuscript can be polished (minor).

Author response: We apologize for this mistake and have revised this manuscript thoroughly for language editing.

6. The authors should cross-check all abbreviations in the manuscript. Initially, define in full name followed by abbreviation.

Author response: We apologize for this mistake and have revised this manuscript thoroughly for abbreviations.

Reviewer #3: This study investigated the commitment status of Syrian population to the protective measures against SARS-CoV-2 viral infection during COVID-19 pandemic. The manuscript is well organized and worth to be published. Yet, given the high contagious nature of the SARS-CoV-2 virus, commitment to the protective measures may only help to delay the infection to some limited extent, and eventually all members of the public will get infected by the virus in the long time. So, the paper will be more attractive if the authors can also provide some insightful understanding about infectious diseases like the SARS-CoV-2 infection, so that the public know what to do to avoid to develop into severe case of COVID-19 if they are infected by SARS-CoV-2 virus.

In talking about infectious diseases, it would be better if we distinguish infections from diseases. Infection is the presence of pathogen, which is not a sufficient condition for disease. [1] Like most of the other viral and bacterial infections, the SARS-COV-2 viral infection is self-limiting [2]. When the human cells are infected by virus or bacteria, the human immunity will actively induce cell self-destruction (programmed cell deaths like necroptosis [3] and pyroptosis [4]) to stop the intracellular infection and reuse the nutrition from the degradation of these destroyed cells and the microorganisms in these cells to rebuild the tissue cells [5]. So by actively destroy damaged somatic cells, the viral and bacterial infections are also removed and become self-limiting [6]. Therefore most of the COVID-19 cases are asymptomatic or very mild. Most of the severe cases of infectious diseases are the result of the overreaction of the strong yet malfunctioning immune system, as Sir William Osler stated more than 100 years ago: "Except on few occasions, the patient appears to die from the body’s response to infection rather than from it. Sir William Osler (1904)" [7]. As the human immune system also plays a virtual role in nutrition acquisition from the degradation of viral damaged epithelial cells, the nutrition surge coupled with the overnutrition state in some patients with obesity or metabolic syndromes may contribute to lipotoxicity and damage in non-adipose tissues, triggering hyperinflammation and contribute to the malfunction of the immune system. So the SARS-CoV-2 viral infection is only the trigger for the COVID-19 disease, and the real cause for the severe cases of COVID-19 disease are autoimmune disorder [8] caused by overnutrition which leads to the hyperinflammation and cytokine storm in these patients.

References:

1. V Humphries DL, Scott ME, Vermund SH (2021) Pathways linking nutritional status and infectious disease: causal and conceptual frameworks. In: Nutrition and infectious diseases, Shifting the Clinical Paradigm (eds DL Humphries, ME Scott, SH Vermund), pp. 3–22. Cham, Switzerland: Humana. DOI: 10.1007/978-3-030-56913-6_1

2. Kumar S, Veldhuis A, Malhotra T (2021) Neuropsychiatric and Cognitive Sequelae of COVID-19. Front Psychol 12:577529. DOI: 10.3389/fpsyg.2021.577529.

3. Nailwal, H., Chan, F.KM. Necroptosis in anti-viral inflammation. Cell Death Differ 26, 4–13 (2019). DOI: 10.1038/s41418-018-0172-x

4. Jorgensen I, Miao EA (2015) Pyroptotic cell death defends against intracellular pathogens. Immunological Reviews. 265 (1): 130-142. DOI:10.1111/imr.12287

5. Maitre Y, Mahalli R, Micheneau P, Delpierre A, Amador G, Denis F (2021) Evidence and Therapeutic Perspectives in the Relationship between the Oral Microbiome and Alzheimer's Disease: A Systematic Review. Int J Environ Res Public Health 18(21):11157. DOI: 10.3390/ijerph182111157.

6. Levin BR, Baquero F, Ankomah P, McCall IC (2017) Phagocytes, Antibiotics, and Self-Limiting Bacterial Infections. Trends in Microbiology, 25(11):878-892. DOI: 10.1016/j.tim.2017.07.005

7. Dobson GP, Biros E, Letson HL and Morris JL (2021) Living in a Hostile World: Inflammation, New Drug Development, and Coronavirus. Front. Immunol. 11:610131. DOI: 10.3389/fimmu.2020.610131

8. Halpert G, Yehuda Shoenfeld Y (2020) SARS-CoV-2, the autoimmune virus. Autoimmunity Reviews 19(12):102695. DOI: 10.1016/j.autrev.2020.102695.

Author response: We really appreciate the efforts made by the reviewer and have used his/her guidance to further improve our manuscript. and we hope that the study is now better than before These amendments have been made to the manuscript text and where these can be viewed (Background section, line 69-76, and Discussion section, line 240-249)

---

## [Decision Letter · Decision Letter 1]

29 Jun 2022

PONE-D-21-39441R1Commitment to protective measures during COVID-19 pandemic in Syria: a nationwide cross-sectional study.PLOS ONE

Dear Dr. Shibani,

Thank you for submitting your manuscript to PLOS ONE. After careful consideration, we feel that it has merit but does not fully meet PLOS ONE’s publication criteria as it currently stands. Therefore, we invite you to submit a revised version of the manuscript that addresses the points raised during the review process. Please address issues raised by the AE and reviewers as suggested in comments for the authors.

We look forward to receiving your revised manuscript.

Kind regards,

Syed Ghulam Sarwar Shah, M.B.B.S., M.A., M.Sc., Ph.D.

Academic Editor

PLOS ONE

Additional Editor Comments:

Thanks for your revised manuscript, which is has following issues that need to be addressed.

Title: Please add ‘the’ before COVID-19 pandemic.

Abstract:

1. Please report the sample size and sample type in the methods section of abstract.

2. Please report the total number of respondents and the response rate in the results section.

3. Please revise this sentence as it is not clear “Statistically significant differences across age, sex, marital status, financial status, employment, and educational attainment when correlated against commitment to preventive measures.”

4. Please reconcile your statements about adherence to preventive measures in the abstract because in the results section you report ‘good adherence ‘while in the conclusion section you state ‘high level of adherence’. Good and High level are not same thing. Moreover, you have reported ‘Low’, ‘Moderate’ and ‘High’ categories of commitment in the methods section (lines 155-156). Therefore, please be consistent and report the level of commitment that was found.

Backgrounds:

1. Could you please report the date on which there were ‘…over 250 million cases and over 5 million deaths (1)’.

2. Please refer to your statement about the possible impact of nutrition and immunity reported in lines 67-76. This information is not relevant because your study is about the commitment and adherence to the preventive measures and nutrition is not one of the measures. Could you please remove this text along with the references cited in this paragraph?

3. Please change ‘6pm-to-6am’to ‘6 pm-to-6 am’ (line 79).

4. Could you please change ‘wearing masks’ to ‘wearing face masks’ (line 83).

5. Could you please report the exact date instead of writing: ‘As of this writing’ in the following sentence ‘As of this writing, 45,468 laboratory-confirmed cases and 2,637 casualties.. (Line 88).

METHODS:

6. Study design, setting, and participants: Please report your total sample size (how many people were invited /approached to complete the survey questionnaire) because there are different numbers reported in the manuscript. For example, in the abstract sample size is ‘10083’ reached out’ while in the methods section you report calculated sample size = 7336 and then you report 8083 people completed the survey and thereafter ‘final sample size of 7,531 participants. Please clearly report your calculated sample size as well as how many people were invited/reached out (if there is difference between these then give reasons for it). Then how many responses were received and how many cases were excluded and final sample size / responses that were analysed. In the abstract you report your final sample size included in the analysis as well as effective response rate.

7. Could you please provide more information about how your sample was ‘nationally representative sample’? (Line 110).

8. You report ‘Chain-referral and convenience sampling methods were employed...’. These might include ‘snowball sampling’? If so, please report it here as well as in the abstract.

9. Please refer to you statement: ‘The questionnaire, based on a previous study,..’ (Line 110). Could you please provide a reference/cite this study that used this questionnaire originally?

10. Could you please report in the methods section how financial status was categorised because Table 5 includes categories of the financial status of the participants.

11. Statistical analysis: Could you please add the abbreviation of names of the researchers who manually entered data in the following sentence: ‘Data from the hard copy questionnaires were entered manually by the investigators (AB, XY, etc) (lines 149-150).

12. ‘A 13 point scale was used to measure the level of commitment (Line 152) was this scale created by the authors of this study or it was already reported in the original questionnaire adopted from earlier studies (which ones?).

13. Please report the lowest and highest scores in the 13-point scale.

14. Ethics & Consent: The authors report that ‘Informed consent was obtained from every participant…’ (line 164). Could you please report how consent from participants was obtained and in which form it was obtained?

RESULTS:

15. Participant characteristics: The authors report that ‘Of 10083 participants, 7531 agreed to participate,’ (Line 170) whereas in the methods section they report that 8083 people completed the survey (Line 123). Please report the actual number of participants who completed the survey and then how many surveys were fully completed. You might like to say: ‘Of 8083 retuned surveys, 7531 surveys were fully completed and analysed’. Then you can report your affective response rate.

16. Participant characteristics: Please avoid reporting same information in the text as well as in Tables. For example: ‘…3505 (46.5%) were males and 4026 (53.5%) were females. 172 The dominant age group was 18-24 years old: 3124 (41.5%), followed by 25 - 44 years old: 2338 173 (31%), 45-65 years old: 1686 (22.4%), and > 65 years: 383 (5.1%).’ (Lines 171-173. Here your repot counts and % of items/variables that are also given in Table 1. Report only the main / major findings in the text and refer to table for numbers and %. For example: Most of the respondents were women and the dominant age group was 18-24 years (Table 1).

17. Please revise this sentence: Over the half of the study population originated from the central governorates: 4376 (58.1%), and 1165 (15.5%) from the western governorates. (Lines 174-175). You might like to report as: Just more than half of the respondents originated from the central governorates while the lowest respondents were from the western governorates.

18. Please revise the text reported in lines 176-181 as suggested above.

19. Please revise the text about the findings about ‘COVID-19 related information’ and ‘Commitment to preventive measures’ as suggested in comments 15-16 above.

20. Please revise ‘…showed poor commitment. (Table 3).’ As ‘...showed low commitment. (Table 3).’ (Line 201).

21. Table 3: In this table commitment levels are reported as ‘Bad, Moderate, and Good’, whereas in the methods section commitment levels reported are: ‘Low’, ‘Moderate’ and ‘High’ (lines 155-156). Please be consistent and report the level of commitment as reported in the methods section.

22. Table 4, Please start each statement with a capital letter i.e. wearing as Wearing, cancelled as Cancelled and so on in this Table.

23. P-values: Please report p-values as <0.001 for all values 0.000 or <0.0001 in the section about ‘Correlations between commitment to preventive measures and participants characteristics’ (lines 212-233), Table 5, Table 6 and elsewhere.

24. Table 5 & Table 6: Please revise ‘Bad’ to ‘Low’ in column ‘Level of commitment’ in these tables and elsewhere.

DISCUSSION

25. Please remove the text in the first paragraph in lines 240-248 “In talking about infectious diseases,…………. also removed and become self-limiting (28).” This information is about issues that are different from the focus of this study which is about the commitment to preventive measures to prevent the spread of COVID-19. Please remove all references (23-27) cited in this para.

26. The authors argue that ‘This level of adherence is similar to that found in a Belgian study’; however, comparison between Belgium and Syria does not seem to be reasonable because the two countries are very different in many aspects social, economic and so on. The authors might like to revise this argument. Similarly in the following sentence the authors compare their findings with the UK and Germany also. The authors should compare their findings with the countries in their region and with similar socio-economic situation.

27. Please change ‘The majority of our sample were young...’ (Line 261) to ‘The majority of our participants were young…’

28. The authors report that ‘Low income and unemployment were correlated with lower adherence to IPC measures, while higher income and gainful employment was correlated with higher adherence’ (Lines 273-274). Table 5 shows similar findings about the low education level and vice versa. The authors could include the findings about the education level in this sentence. These findings are important from the policy perspective and could be included/highlighted in the conclusion.

CONCLUSIONS

29. Conclusion reported in the abstract and in the main manuscript should convey the same message so please revise your conclusion at both places.

Reviewers' comments:

Reviewer's Responses to Questions

**Comments to the Author**

1. If the authors have adequately addressed your comments raised in a previous round of review and you feel that this manuscript is now acceptable for publication, you may indicate that here to bypass the “Comments to the Author” section, enter your conflict of interest statement in the “Confidential to Editor” section, and submit your "Accept" recommendation.

Reviewer #1: All comments have been addressed

Reviewer #2: All comments have been addressed

Reviewer #3: All comments have been addressed

Reviewer #4: All comments have been addressed

2. Is the manuscript technically sound, and do the data support the conclusions?

Reviewer #1: Yes

Reviewer #2: Yes

Reviewer #3: Yes

Reviewer #4: Yes

3. Has the statistical analysis been performed appropriately and rigorously? 

Reviewer #1: Yes

Reviewer #2: Yes

Reviewer #3: Yes

Reviewer #4: Yes

4. Have the authors made all data underlying the findings in their manuscript fully available?

Reviewer #1: Yes

Reviewer #2: Yes

Reviewer #3: Yes

Reviewer #4: Yes

5. Is the manuscript presented in an intelligible fashion and written in standard English?

Reviewer #1: Yes

Reviewer #2: Yes

Reviewer #3: Yes

Reviewer #4: Yes

6. Review Comments to the Author

Reviewer #1: In this paper, the authors evaluate how the Syrian population adheres to these measures and analyze the relationship between demographic variables and adherence. The manuscript is well written and intelligently presented. In addition, the study is nicely designed. I congratulate the authors on this work. The manuscript is suitable for publication.

Reviewer #2: This manuscript entitled "Commitment to protective measures during COVID-19 pandemic in Syria: a nationwide cross-sectional study" has improved.

Reviewer #3: As the authors have addressed all my concerns in my previous review comments, the manuscript can be accepted for publication in its present form.

Reviewer #4: Thank you for the opportunity to review this manuscript. It is an interesting and very well written and structured work that looks at public compliance with preventive measures in Syria during the COVID-19 pandemic. The results are interesting and informative. I think that the authors have successfully addressed the concerns and provided an important addition to the literature. My only suggestion would be to maybe add absolute as well as relative frequencies to Figure 1 to more easily identify the distribution based on sample size.

7. PLOS authors have the option to publish the peer review history of their article (what does this mean?). If published, this will include your full peer review and any attached files.

Reviewer #1: No

Reviewer #2: **Yes: **Vinay Kumar

Reviewer #3: **Yes: **Ligen Yu

Reviewer #4: No

---

## [Author Response · Author response to Decision Letter 1]

20 Jul 2022

Response letter

 We would like to thank the editor for taking the valuable time to review this manuscript. We have thoroughly assessed the comments and implemented them into our revised manuscript, now that this article has been made suitable, we aspire to have this article published in your journal. 

The following letter will report the responses to the comments.

Editor comments:

- Title: Please add ‘the’ before COVID-19 pandemic.

Author response: We thank the editor for the guidance, and have made the asked amendment

Abstract:

1- Please report the sample size and sample type in the methods section of abstract.

Author response: We thank the editor for the guidance, and have made the asked amendment. These amendments have been made to the manuscript text and where these can be viewed (Abstract section, line 34-35)

2- Please report the total number of respondents and the response rate in the results section.

Author response: the total number of respondents and the response rate have been added in the results section, which can be viewed in (Abstract section, line 38)

3- Please revise this sentence as it is not clear “Statistically significant differences across age, sex, marital status, financial status, employment, and educational attainment when correlated against commitment to preventive measures.”

Author response: We thank the editor for the guidance, and have revised this sentence and made it more clear. This can be viewed (Abstract, line 41-44)

4- Please reconcile your statements about adherence to preventive measures in the abstract because in the results section you report ‘good adherence ‘while in the conclusion section you state ‘high level of adherence’. Good and High level are not same thing. Moreover, you have reported ‘Low’, ‘Moderate’ and ‘High’ categories of commitment in the methods section (lines 155-156). Therefore, please be consistent and report the level of commitment that was found.

Author response: We understand the editor’s point, and have revised the manuscript to ensure we have used the same term and classification in all sections, and correction to the mentioned sentence were made. these can be viewed (Abstract, line 40 -41)

Background:

1- Could you please report the date on which there were ‘…over 250 million cases and over 5 million deaths (1)’

Author response: We apologize for this mistake, and have reported the date. These amendments have been made to the manuscript text and where these can be viewed (Background, line 57-58)

2- Please refer to your statement about the possible impact of nutrition and immunity reported in lines 67-76. This information is not relevant because your study is about the commitment and adherence to the preventive measures and nutrition is not one of the measures. Could you please remove this text along with the references cited in this paragraph?

Author response: We agree with the editor, and have removed this paragraph from the manuscript. These amendments have been made to the manuscript text and where these can be viewed in the track change manuscript (Background, line 70-79)

3- Please change ‘6pm-to-6am’to ‘6 pm-to-6 am’ (line 79) 

Author response: We apologize for this mistake and have made the asked modifications. (Background, line 82)

4- Could you please change ‘wearing masks’ to ‘wearing face masks’ (line 83).

Author response: We apologize for this mistake and have made the asked modifications. (Background, line 87)

5- Could you please report the exact date instead of writing: ‘As of this writing’ in the following sentence ‘As of this writing, 45,468 laboratory-confirmed cases and 2,637 casualties.. (Line 88).

Author response: We understand the editor’s point, and have mentioned the exact date for these figures. These amendments have been made to the manuscript text and where these can be viewed (Background section, line 91-92)

METHODS:

6- Study design, setting, and participants: Please report your total sample size (how many people were invited /approached to complete the survey questionnaire) because there are different numbers reported in the manuscript. For example, in the abstract sample size is ‘10083’ reached out’ while in the methods section you report calculated sample size = 7336 and then you report 8083 people completed the survey and thereafter ‘final sample size of 7,531 participants. Please clearly report your calculated sample size as well as how many people were invited/reached out (if there is difference between these then give reasons for it). Then how many responses were received and how many cases were excluded and final sample size / responses that were analysed. In the abstract you report your final sample size included in the analysis as well as effective response rate.

Author response: We understand the editor’s point and would like to clarify that initially the sample size calculation software suggested a minimum sample size of 7336. In order to achieve this goal, we reached out to 10083 individuals, and of these only 8083 responded back to us. After reviewing these responses we excluded those who did not meet our inclusion criteria, which eventually gave us a final figure of 7531. This clarification was made to the manuscript where it can be viewed (Method section, line 124- 132)

7- Could you please provide more information about how your sample was ‘nationally representative sample’? (Line 110).

Author response: We thank the editor for this comment, and have removed this statement from the methods section as it is fully explained later on the discussion section. These amendments have been made to the manuscript text and where these can be viewed (Method section, line 113)

8- You report ‘Chain-referral and convenience sampling methods were employed...’. These might include ‘snowball sampling’? If so, please report it here as well as in the abstract.

Author response: We agree with the editor, and yes indeed, snowball sampling is a synonym for chain-referral sampling. We added this term appropriately in the method section where these can be viewed (methods section, line 116) 

9- Please refer to you statement: ‘The questionnaire, based on a previous study,..’ (Line 110). Could you please provide a reference/cite this study that used this questionnaire originally?

Author response: We thank the editor for the guidance. However, the questionnaire we used is a newly developed one based on multiple different questionnaires from multiple previous studies (references 15-17). We modified this sentence and made this idea clear which can be viewed in (Method section, line 113-114)

10- Could you please report in the methods section how financial status was categorised because Table 5 includes categories of the financial status of the participants

Author response: We thank the editor for the guidance. We mentioned the categories in the method section as required (Methods section, line139-140)

11- Statistical analysis: Could you please add the abbreviation of names of the researchers who manually entered data in the following sentence: ‘Data from the hard copy questionnaires were entered manually by the investigators (AB, XY, etc) (lines 149-150).

Author response: We agree with the editor, and have added the initials of the authors who manually entered the data. These amendments have been made to the manuscript text and where these can be viewed (Methods section, line 155-156).

12- ‘A 13 point scale was used to measure the level of commitment (Line 152) was this scale created by the authors of this study or it was already reported in the original questionnaire adopted from earlier studies (which ones?).

Author response: We understand the editor’s point and would like to clarify that the scale was developed by the investigators for this study. We also clarified this point in the manuscript. These amendments have been made to the manuscript text and where these can be viewed (Method section, line 158-159) 

13- Please report the lowest and highest scores in the 13-point scale.

Author response: We thank the editor for the guidance. We added the lowest and highest scores in the 13-point scale (Methods section, line160)

14- Ethics & Consent: The authors report that ‘Informed consent was obtained from every participant…’ (line 164). Could you please report how consent from participants was obtained and in which form it was obtained?

Author response: We thank the editor for the guidance. We clarified that a written informed consent was obtained from every participant as each questionnaire had an informed consent form (the first page in the hard copy version and the first question in the digital one) needs to be signed by the respondents prior to participation. (Methods section, line171 - 173)

RESULTS:

15- Participant characteristics: The authors report that ‘Of 10083 participants, 7531 agreed to participate,’ (Line 170) whereas in the methods section they report that 8083 people completed the survey (Line 123). Please report the actual number of participants who completed the survey and then how many surveys were fully completed. You might like to say: ‘Of 8083 retuned surveys, 7531 surveys were fully completed and analysed’. Then you can report your affective response rate

Author response: We thank the editor for the guidance. We addressed this point and made it clear in both results and Methods sections. (Methods section, line1791 - 181)

16. Participant characteristics: Please avoid reporting same information in the text as well as in Tables. For example: ‘…3505 (46.5%) were males and 4026 (53.5%) were females. 172 The dominant age group was 18-24 years old: 3124 (41.5%), followed by 25 - 44 years old: 2338 173 (31%), 45-65 years old: 1686 (22.4%), and > 65 years: 383 (5.1%).’ (Lines 171-173. Here your repot counts and % of items/variables that are also given in Table 1. Report only the main / major findings in the text and refer to table for numbers and %. For example: Most of the respondents were women and the dominant age group was 18-24 years (Table 1).

17. Please revise this sentence: Over the half of the study population originated from the central governorates: 4376 (58.1%), and 1165 (15.5%) from the western governorates. (Lines 174-175). You might like to report as: Just more than half of the respondents originated from the central governorates while the lowest respondents were from the western governorates.

18. Please revise the text reported in lines 176-181 as suggested above.

19. Please revise the text about the findings about ‘COVID-19 related information’ and ‘Commitment to preventive measures’ as suggested in comments 15-16 above.

Author response: We thank the editor for the guidance. We addressed this point were appropriate across all result sections. 

20. Please revise ‘…showed poor commitment. (Table 3).’ As ‘...showed low commitment. (Table 3).’ (Line 201).

Author response: We thank the editor for the guidance. We addressed this issue across all result sections. 

21. Table 3: In this table commitment levels are reported as ‘Bad, Moderate, and Good’, whereas in the methods section commitment levels reported are: ‘Low’, ‘Moderate’ and ‘High’ (lines 155-156). Please be consistent and report the level of commitment as reported in the methods section.

Author response: We thank the editor for the guidance. We addressed this issue across all result sections.

22. Table 4, Please start each statement with a capital letter i.e. wearing as Wearing, cancelled as Cancelled and so on in this Table.

Author response: We thank the editor for the guidance. We addressed this issue and corrected all statements.

23. P-values: Please report p-values as <0.001 for all values 0.000 or <0.0001 in the section about ‘Correlations between commitment to preventive measures and participants characteristics’ (lines 212-233), Table 5, Table 6 and elsewhere.

Author response: We thank the editor for the guidance. We addressed this issue across the manuscript

24. Table 5 & Table 6: Please revise ‘Bad’ to ‘Low’ in column ‘Level of commitment’ in these tables and elsewhere.

Author response: We thank the editor for the guidance. We addressed this issue across the manuscript

DISCUSSION

25. Please remove the text in the first paragraph in lines 240-248 “In talking about infectious diseases,…………. also removed and become self-limiting (28).” This information is about issues that are different from the focus of this study which is about the commitment to preventive measures to prevent the spread of COVID-19. Please remove all references (23-27) cited in this para.

Author response: We agree with editor’s point of view and have removed this paragraph and the corresponding references. 

26. The authors argue that ‘This level of adherence is similar to that found in a Belgian study’; however, comparison between Belgium and Syria does not seem to be reasonable because the two countries are very different in many aspects social, economic and so on. The authors might like to revise this argument. Similarly in the following sentence the authors compare their findings with the UK and Germany also. The authors should compare their findings with the countries in their region and with similar socio-economic situation.

Author response: We understand the editor’s point of view and we would like to clarify that we compared our results with results from countries with different ranking because we believe that it is important to understand our country’s situation compared to the world.

27. Please change ‘The majority of our sample were young...’ (Line 261) to ‘The majority of our participants were young…’

Author response: We thank the editor for the guidance and have made the required modification (Discussion section, line 273)

28. The authors report that ‘Low income and unemployment were correlated with lower adherence to IPC measures, while higher income and gainful employment was correlated with higher adherence’ (Lines 273-274). Table 5 shows similar findings about the low education level and vice versa. The authors could include the findings about the education level in this sentence. These findings are important from the policy perspective and could be included/highlighted in the conclusion.

Author response: We thank the editor for the guidance. However, we talked about the correlation between educational level and commitment to protective measures later in the discussion with its own paragraph (Discussion section, line 303-312)

CONCLUSIONS

29. Conclusion reported in the abstract and in the main manuscript should convey the same message so please revise your conclusion at both places.

Author response: We thank the editor for the guidance. We modified the conclusion in the abstract to convey the main message as the main conclusion.

---

## [Editor Report · Decision Letter 2]

4 Aug 2022

PONE-D-21-39441R2Commitment to protective measures during the COVID-19 pandemic in Syria: a nationwide cross-sectional study.PLOS ONE

Dear Dr. Shibani,

Thank you for submitting your manuscript to PLOS ONE. After careful consideration, we feel that it has merit but does not fully meet PLOS ONE’s publication criteria as it currently stands. Therefore, we invite you to submit a revised version of the manuscript that addresses the points raised during the review process. Please submit your revised manuscript by Sep 18 2022 11:59PM. If you will need more time than this to complete your revisions, please reply to this message or contact the journal office at plosone@plos.org. Please include the following items when submitting your revised manuscript:A rebuttal letter that responds to each point raised by the academic editor and reviewer(s). You should upload this letter as a separate file labeled 'Response to Reviewers'.A marked-up copy of your manuscript that highlights changes made to the original version. You should upload this as a separate file labeled 'Revised Manuscript with Track Changes'.An unmarked version of your revised paper without tracked changes. You should upload this as a separate file labeled 'Manuscript'.If applicable, we recommend that you deposit your laboratory protocols in protocols.io to enhance the reproducibility of your results. Protocols.io assigns your protocol its own identifier (DOI) so that it can be cited independently in the future. For instructions see: https://journals.plos.org/plosone/s/submission-guidelines#loc-laboratory-protocols. Additionally, PLOS ONE offers an option for publishing peer-reviewed Lab Protocol articles, which describe protocols hosted on protocols.io. Read more information on sharing protocols at https://plos.org/protocols?utm_medium=editorial-email&utm_source=authorletters&utm_campaign=protocols.

We look forward to receiving your revised manuscript.

Kind regards,

Syed Ghulam Sarwar Shah, M.B.B.S., M.A., M.Sc., Ph.D.

Academic Editor

PLOS ONE

Journal Requirements:

Additional Editor Comments:

Comments from the Academic Editor:

Many thanks for addressing issues raised by AE. However, there are still few issues that need to be addressed as suggested below.

1. Authorship: A new author named ‘Dana Alakhrass’ with affiliation no. 6 has been added in this version of the manuscript. This person was not a author in the original and revised submission R1. This person could not be added as an author at this stage. If this person helped in the latest revision of the manuscript, then their name could be reported in the acknowledgments. Please refer to the PLOS ONE Guidelines for the authorship for further guidance on the issue.

2. Abstract; Conclusion- Please change 'a nationwide awareness campaigns ' to either 'A nationwide awareness campaign' or 'Nationwide awareness campaigns' in the conclusion in the abstract.

3. Background: Lines 58-59. Please correct the following sentence 'to the date of writing in December 2021'. It could be ‘as of XX December 2021' (put a particular date instead of to the date of writing).

4. Methods: Lines 115-116, Please change from 'based on a previous studies' to 'based on previous studies'

5. Financial status in Table 1 and other text: Please change financial status category of 'bad' to 'low', which reads better. Make this change in all places where you have reported 'bad' financial status.

6. Table 4: Specific preventive measures: Some of these statements start with a past tense while other use different tense. Could you please correct these statements so that they are in the same tense? All these statements should start with a capital letter.

7. Table 4: What do you mean by the amount in the following sentence 'Reduce the amount you go to shops'? Is it the 'time' or 'number of times' or something else? Please change accordingly.

8. Legends of figures and table captions: Please put full stop after the number. Change from Table .1 to Table 1. Please do the same for all figures and tables.

9. References: please update your references and include the vol. issue and page numbers as well as DOI for all journal articles and URLs for internet / website sources. In some reference year is written twice, please report it only once.
---

## [Author Response · Author response to Decision Letter 2]

7 Sep 2022

We would like to thank the editors for taking their valuable time to review this manuscript. We have thoroughly assessed the comments and implemented them into our revised manuscript, now that this article has been made suitable we aspire to have this article published in your journal. 

The following letter will report the responses to the reviewers’ comments.

Comments from the Academic Editor:

1. Authorship: A new author named ‘Dana Alakhrass’ with affiliation no. 6 has been added in this version of the manuscript. This person was not a author in the original and revised submission R1. This person could not be added as an author at this stage. If this person helped in the latest revision of the manuscript, then their name could be reported in the acknowledgments. Please refer to the PLOS ONE Guidelines for the authorship for further guidance on the issue.

Author response: We apologize for this mistake and have removed this person name from the authorship list to the acknowledgment section These amendments have been made to the manuscript text and where these can be viewed (acknowledgment section, lines 332-333)

2. Abstract; Conclusion- Please change 'a nationwide awareness campaigns ' to either 'A nationwide awareness campaign' or 'Nationwide awareness campaigns' in the conclusion in the abstract. 

Author response: We thank the academic Editor for the guidance, and have changed 'a nationwide awareness campaigns ' to 'Nationwide awareness campaigns'. These amendments have been made to the manuscript text and where these can be viewed (Abstract section, lines 48-49)

3. Background: Lines 58-59. Please correct the following sentence 'to the date of writing in December 2021'. It could be ‘as of XX December 2021' (put a particular date instead of to the date of writing).

Author response: We thank the AE for the comment, and have added the exact date corresponding with the numbers to the manuscript. These amendments have been made to the manuscript text and where these can be viewed (Introduction section, lines 56-57)

4. Methods: Lines 115-116, Please change from 'based on a previous studies' to 'based on previous studies'

Author response: We apologize for this mistake and have revised the sentence. These amendments have been made to the manuscript text and where these can be viewed (methods section, line104)

5. Financial status in Table 1 and other text: Please change financial status category of 'bad' to 'low', which reads better. Make this change in all places where you have reported 'bad' financial status.

Author response: We thank the AE for the comment, and have changed the financial status category of 'bad' to 'low' throughout the manuscript.

6. Table 4: Specific preventive measures: Some of these statements start with a past tense while other use different tense. Could you please correct these statements so that they are in the same tense? All these statements should start with a capital letter.

Author response: We thank the academic Editor for the guidance, and have changed the tenses of all statements to be in the past tense. These amendments have been made to the manuscript text and where these can be viewed (Table 4)

7. Table 4: What do you mean by the amount in the following sentence 'Reduce the amount you go to shops'? Is it the 'time' or 'number of times' or something else? Please change accordingly.

Author response: We apologize for this mistake and have clarified the exact meaning of the question by changing 'Reduce the amount you go to shops'? to 'Reduce the number of times you go to shops'? . These amendments have been made to the manuscript text and where these can be viewed (Table 4.)

8. Legends of figures and table captions: Please put full stop after the number. Change from Table .1 to Table 1. Please do the same for all figures and tables.

Author response: We apologize for this mistake and have applied the required amendments. (Tables and figures legends section, lines 335-342)

9. References: please update your references and include the vol. issue and page numbers as well as DOI for all journal articles and URLs for internet / website sources. In some reference year is written twice, please report it only once.

Author response: We have updated the references list to include the vol. issue and page numbers as well as DOI for all journal articles and URLs for internet / website sources.

---

## [Editor Report · Decision Letter 3]

21 Sep 2022

Commitment to protective measures during the COVID-19 pandemic in Syria: a nationwide cross-sectional study.

PONE-D-21-39441R3

Dear Dr. Shibani,

We’re pleased to inform you that your manuscript has been judged scientifically suitable for publication and will be formally accepted for publication once it meets all outstanding technical requirements.

Kind regards,

Syed Ghulam Sarwar Shah, M.B.B.S., M.A., M.Sc., Ph.D.

Academic Editor

PLOS ONE

Additional Editor Comments (optional):

Thank you for addressing all issues raised by the AE and reviewers in your revised manuscript.
---

## [Editor Report · Acceptance letter]

5 Oct 2022

PONE-D-21-39441R3 

Commitment to protective measures during the COVID-19 pandemic in Syria: a nationwide cross-sectional study. 

Dear Dr. Shibani:

I'm pleased to inform you that your manuscript has been deemed suitable for publication in PLOS ONE. Congratulations! Your manuscript is now with our production department. 

Kind regards, 

on behalf of

Dr. Syed Ghulam Sarwar Shah 

Academic Editor

PLOS ONE